# Effectiveness of repeated mutagenesis of sesame crosses for enhancing polygenic variability in $F_2M_2$ generation

**Rajesh Kumar Kar[1], Tapash Kumar Mishra[2], Banshidhar Pradhan[2], Ahmed Gaber[3], Dibyabharati Sahu[1], Subhashree Das[2], Deepak Kumar Swain[4], Srikrushna Behera[5], Aditya Kiran Padhiary[6], Sarthak Pattanayak[7], S. P. Monalisa[8], Ritu Kumari Pandey[9], Poonam Preeti Pradhan[10], Debendra Nath Sarangi[11], Mihir Ranjan Mohanty[12], Biswajit Lenka[1], Lingaraj Dip[6], Anannya Jena[6], Uma Pradhan[6], Siba Prasad Mishra[13], Manas Kumar Patel[14], Rashmi Prabha Mishra[15], Akbar Hossain[16]***

1 Department of Genetics and Plant Breeding, Institute of Agricultural Sciences (IAS), Siksha 'O' Anusandhan (Deemed to be University), Bhubaneswar, Odisha, India, 2 Department of Genetics and Plant Breeding, College of Agriculture, Odisha University of Agriculture and Technology (OUAT), Bhubaneswar, Odisha, India, 3 Department of Biology, College of Science, Taif University, Taif, Saudi Arabia, 4 Department of Agricultural Statistics, Institute of Agricultural Sciences (IAS), Siksha 'O' Anusandhan (Deemed to be University), Bhubaneswar, Odisha, India, 5 KrishiVigyan Kendra (KVK), Odisha University of Agriculture and Technology (OUAT), Bhawanipatna, Odisha, India, 6 KrishiVigyan Kendra (KVK), Odisha University of Agriculture and Technology (OUAT), Sambalpur, Odisha, India, 7 KrishiVigyan Kendra (KVK), Odisha University of Agriculture and Technology (OUAT), Balangir, Odisha, India, 8 Department of Seed Science and Technology, Institute of Agricultural Sciences (IAS), Siksha 'O' Anusandhan (Deemed to be University), Bhubaneswar, Odisha, India, 9 Directorate of Plant Protection, Quarantine and Storage, Central Integrated Pest Management Centre, Bhubaneswar, Odisha, India, 10 Department of Soil Sciences and Agricultural Chemistry, Institute of Agricultural Sciences (IAS), Siksha 'O' Anusandhan (Deemed to be University), Bhubaneswar, Odisha, India, 11 ICAR-Central Institute for Women in Agriculture (ICAR-CIWA), Bhubaneswar, Odisha, India, 12 Regional Research and Technology Transfer Sub-Station (RRTTSS), Odisha University of Agriculture and Technology (OUAT), Jeypore, Odisha, India, 13 KrishiVigyan Kendra (KVK), Odisha University of Agriculture and Technology (OUAT), Jajpur, Odisha, India, 14 Department of Fruit Science, Institute of Agricultural Sciences (IAS), Siksha 'O' Anusandhan (Deemed to be University), Bhubaneswar, Odisha, India, 15 KrishiVigyan Kendra (KVK), Odisha University of Agriculture and Technology (OUAT), Angul, Odisha, India, 16 Division of Soil Science, Bangladesh Wheat and Maize Research Institute, Nashipur, Dinajpur, Bangladesh

* akbarhossainwrc@gmail.com

**Data Availability Statement:** All relevant data are within the paper and its Supporting information files.

## Abstract

The value of combining hybridization and mutagenesis in sesame was examined to determine if treating hybrid sesame plant material with mutagens generated greater genetic variability in four key productivity traits than either the separate hybridization or mutation of plant material. In a randomized block design with three replications, six $F_2M_2$ varieties, three $F_2$ varieties, and three parental varieties were assessed at Odisha University of Agriculture and Technology, Bhubaneswar, Odisha, India. The plant characteristics height, number of seed capsules per plant, and seed yield per plant had greater variability in the $F_2M_2$ generation than their respective controls ($F_2$), however, the number of primary branches per plant varied less than in the control population. The chances for trait selection to be operative were high for all the characteristics examined except the number of primary branches per plant, as indicated by heritability estimates. Increases in the mean and variability of the characteristics examined indicted a greater incidence of beneficial mutations and the breakdown

**Funding:** This research work was supported by the Department of Genetics and Plant Breeding, College of Agriculture, Odisha University of Agriculture and Technology, Bhubaneswar-751003, Odisha, India. The study was also partially funded by the Taif University Researchers Supporting Project number (TURSP - 2020/39), Taif University, Taif, Saudi Arabia. The funders had no role in study design, data collection and analysis, decision to publish, or preparation of the manuscript.

**Competing interests:** The authors have declared that no competing interests exist.

of undesirable linkages with increased recombination. At both phenotypic and genotypic levels strong positive correlations between both primary branch number and capsule number with seed yield suggest that these traits are important for indirect improvement in sesame seed yield. As a result of the association analysis, sesame seed yield and its component traits improved significantly, which may be attributed to the independent polygenic mutations and enlarged recombination of the polygenes controlling the examined characteristics. Compared to the corresponding control treatment or to one cycle of mutagenic treatment, two cycles of mutagenic treatment resulted in increased variability, higher transgressive segregates, PTS mean and average transgression for sesame seed yield. These findings highlight the value of implementing two EMS treatment cycles to generate improved sesame lines. Furthermore, the extra variability created through hybridization may have potential in subsequent breeding research and improved seed yield segregants may be further advanced to develop ever-superior sesame varieties.

## 1. Introduction

Sesame (*Sesamum indicum* L.) is primarily produced commercially for its edible seed, oil, and flavour. Sesame seeds contain 40–58% oil, 20–25% protein, and 13.5% carbohydrate [1]. A 100 g sample of sesame seeds contains the following nutrients: 49.7 g of fat, 9.85 g of carbohydrates, 17.6 g of protein, 14.9 g of fibre, and approximately 4.48 g of ash [2]. Sesame seeds also include sesamolin and sesamin which are natural preservatives of sesame food products: sesamin is an antioxidant with anti-inflammatory properties [3]. As well, sesame contains significant amounts of high-quality protein with a balanced amino acid profile [4]. Sesame has not been the subject of much research by national or international agricultural research organizations, and as a result research into this crop has lagged significantly behind other important oil seed crops.

Sesame is grown widely in several countries, including China, India, Ethiopia, Uganda, Nigeria, and Myanmar. In India the crop is grown on 1.52 million hectares, with a production of 0.66 million metric tons and productivity of 433 kg/ha, compared to the global average of 512 kg/ha [5]. There is potential to improve from the existing low yield productivity in India.

The commercial production of sesame is constrained by a lack of improved, modern cultivars; existing varieties are susceptible to disease, establish poorly, have unpredictable growth, ripen unevenly, and are at risk of capsule shattering and profuse branching which result in low harvest indices [6]. Sesame improvement has been explored using a variety of breeding techniques; however, these have only resulted in minor increases in productivity. The natural genetic variability of sesame has been depleted by extensive inbreeding types over a long period of time [7].

Recombination and mutation breeding have been identified as pathways by which genetic variability may be increased to facilitate the selection of outstanding genotypes: the effectiveness of these techniques has previously been demonstrated in other crop plants. Mutation breeding is tool for rapidly generating high quality outcomes relatively quickly. Nonetheless, due to their narrow genetic base, the mutation in homozygous genotypes has had limited success. When subjected to mutagenesis, the homozygous genetic material rarely yields favourable mutants due to low mutability. To find such economic mutants in the offspring, it is necessary to evaluate large populations using effective screening procedures. Recombination breeding is a popular technique for crop improvement that produces new and enhanced recombinants by

combining hybridization among parents, followed by selection in subsequent generations. This makes it easier to combine advantageous traits into one genotype. However the potential of recombination in sesame is constrained by correlations between favourable and negative traits: the ability to identify improved genotypes is reduced because the necessary amount of variability between traits does not exist.

Recombination in combination with mutant breeding has been proposed as an innovative strategy to overcome the constraints of traditional breeding practices in sesame. The heterozygosity of the hybrids provides the mutagen a wide genetic foundation to work on, generating variability and facilitating additional selection. Mutagenesis of the hybrid species induces additional micro-mutations [8], enhances recombination of different traits [9], breaks undesirable linkages [10], and creates greater variability than the sum of the variances of mutagenized genotypes and segregated hybrid progenies separately [11, 12]. The potential for enhanced variability has been demonstrated following hybrid mutagenic treatments and improved cultivars have been produced in several crops using this technique. However, research on this approach in sesame is limited.

This paper reports on an analysis conducted to evaluate the outcome in sesame of hybridization with both non-recurrent and recurrent mutagenic treatments on the variability of important quantitative traits related to yield during the $F_2M_2$ generation. Three inter-varietal crosses were made between three adapted sesame varieties (Nirmala, Prachi, and Amrit). These research findings provide insights into the usefulness of hybridization alone and hybridization combined with recurrent and non-recurrent EMS mutagenesis on the potential to increase sesame yield. The potential to identify any additional variability created through hybridization would inform future sesame breeding activities and the selection of superior lines that, after rigorous testing, may be developed into releasable varieties.

## 2. Materials and methods

### 2.1. Selection of sesame parent varieties

Three released varieties of sesame, Nirmala, Prachi, and Amrit, which are recommended for the state of Odisha in eastern India were used as potential parent varieties for this research. Important characteristics of these cultivars are outlined in Table 1.

### 2.2. Experimental location

A field study was conducted at the Economic Botany-II, Genetics and Plant Breeding Department, College of Agriculture, OUAT Bhubaneswar, India.

**Table 1. Place of origin, pedigree, traits of three parental sesame varieties.**

| Variety | Place of origin | Pedigree | Plant type | Maturity duration (days) | Capsule type | Seed colour | Oil content (%) | Average yield (q/ha) |
|---------|-----------------|----------|------------|--------------------------|--------------|-------------|------------------|----------------------|
| Nirmala | OUAT[1] | Mutant of B-67 | Basal branching | 80–85 | Small, and hairy | Grey white | 42–44 | 7.00–7.50 |
| Prachi | OUAT | Mutant of B-67 | Profuse branching | 85–90 (kharif/monsoon) 75–80 (rabi/summer) | Narrow, oblong and less hairy | Black | 42–45 | 9.00–10.00 |
| Amrit | OUAT | XU-2 × Krishna | Profuse branching | 82–85 | Glabrous, and compact | Light Brown | 43–46 | 7.50–8.50 |

[1] OUAT: Odisha University of Agriculture and Technology, in Bhubaneswar, Odisha, India

## 2.3. Sesame hybridization and cross development

To generate different cross combinations (Nirmala x Prachi, Nirmala x Amrit, and Prachi x Amrit), the parental varieties were grown at staggered intervals of seven days. Each crop was grown using agronomic practices recommended within Odisha. To achieve the cross combinations when the corolla of the intended male parent was about to open the plant was emasculated by removing the flower's epipetalous corolla and applying a speck of fevicol to it [13]. Fevicol was also sprayed on the flower's top to prevent the corolla from opening any further. All of these processes were done in the late afternoon. Among the three varieties, three crosses were achieved, excluding reciprocals.

## 2.4. Chemical mutagen selection

Mutations in genes are a common result of chemical mutagens [14, 15]. The most potent and effective chemical mutagen is ethyl methane sulfonate (EMS) [16]. Previous research has demonstrated the usefulness of EMS to induce beneficial mutations in sesame, and in particular that lower concentrations of EMS have the highest mutagenic efficiency and effectiveness in inducing beneficial mutations [17–23]. Accordingly, we selected a dosage rate of 0.5% EMS (for three hours per cycle) for the treatment of sesame seed materials in this research.

## 2.5. Sesame field experiment

The produced $F_1$ hybrids were subjected to two cycles of ethyl methane sulfonate treatment, at a dose of 0.5% per cycle for three hours. The number of treated and crossed populations (excluding parents) increased from 6 in the $F_1M_1$ to 9 in the $F_2M_2$ generation, when the generations were advanced from $F_1M_1$ to $F_2M_2$ with and without recurring chemical mutagenesis (Fig 1).

The nine populations that made up the material for the $F_1M_1$ generation included the untreated $F_{1s}$, the parental varieties, and the EMS-treated $F_{1s}$ (classified as $F_1{}^1M_{1s}$) from the crosses Nirmala x Prachi, Nirmala x Amrit, and Prachi x Amrit. Using bulk seeds from the sample plants, the materials from the $F_1M_1$ generation were advanced to the $F_2M_2$.

The treated populations, i.e., $F_1M_{1s}$, were advanced to the next generation with and without recurrent mutagenic treatment. Three additional populations arose from two cycles of EMS treatment, and the remaining six populations were the same as in $F_1M_1$ except for the advancement of the generation to $F_2M_2$ and $F_2$. Therefore, in the $F_2M_2$ generation, there were 12

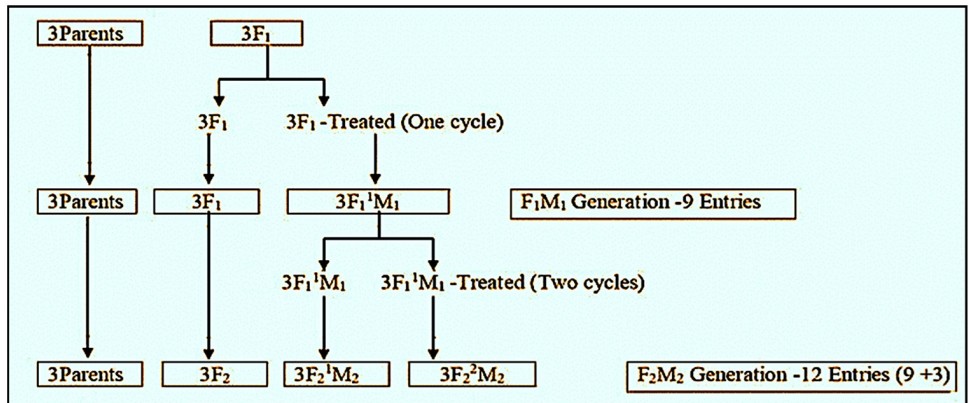

**Fig 1. Overview of sesame recurrent mutagenesis in the field experiment.**

**Table 2. Sesame $F_2M_2$ generation treatment symbols and descriptions of the treatments.**

| Sl. No. | Treatments | Description |
|---|---|---|
| 1. | $V_1$ | 1st Variety (Nirmala) |
| 2. | $V_2$ | 2nd Variety (Prachi) |
| 3. | $V_3$ | 3rd Variety (Amrit) |
| 4. | $^1F_2$ | Cross 1's (Nirmala x Prachi) $F_2$ generation |
| 5. | $^2F_2$ | Cross 2's (Nirmala x Amrit) $F_2$ generation |
| 6. | $^3F_2$ | Cross 3's (Prachi x Amrit) $F_2$ generation |
| 7. | $^1F_2{}^1M_2$ | $F_2M_2$ generation after one cycle of EMS treatment of $F_1$ seeds from cross 1 |
| 8. | $^2F_2{}^1M_2$ | $F_2M_2$ generation after one cycle of EMS treatment of $F_1$ seeds from cross 2 |
| 9. | $^3F_2{}^1M_2$ | $F_2M_2$ generation after one cycle of EMS treatment of $F_1$ seeds from cross 3 |
| 10. | $^1F_2{}^2M_2$ | $F_2M_2$ generation from $F_2{}^1M_2$ seeds* of cross 1 after two cycles of EMS treatment |
| 11. | $^2F_2{}^2M_2$ | $F_2M_2$ generation from $F_2{}^1M_2$ seeds* of cross 2 after two cycles of EMS treatment |
| 12. | $^3F_2{}^2M_2$ | $F_2M_2$ generation from $F_2{}^1M_2$ seeds* of cross 3 after two cycles of EMS treatment |

*Mark denotes $F_1{}^1M_1$ plant seeds. The cross number is represented by the superscript $F_2$ in the treatment symbol rather than a filial generation; similarly, the superscript $M_2$ denotes cycles that have been subjected to mutagens; EMS, ethyl methane sulfonate.

populations, i.e., three $F_2{}^2M_2$, three $F_2{}^1M_{2s}$ three $F_{2s}$ (controls), and three parents. Treatment symbols and descriptions for the $F_2M_2$ generation treatments, including controls, are shown in Table 2.

To grow plants of the six $F_2M_{2s}$, three $F_{2s}$, and three parental sesame types, a randomized complete block design with three replications was used. Sesame seed was sown in plots of 8 rows, each 2.5 meters long, with 30 cm x 10 cm spacing. After dibbing two to three seeds per hill, subsequent thinning left only one seedling per hill for a good crop stand. Other crop management (fertilizer application, watering, and other cultivation activities) was undertaken following the standard recommended practice for sesame cultivation in Odisha recommended by Odisha State Government.

## 2.6. Data sampling

Thirty plants were randomly selected from each treatment plot. From each plant height, number of primary branches per plant, the number of capsules per plant, and yield were all recorded.

## 2.7. Statistical evaluation

Using the means of each of the four measured characteristics (plant height, number of primary branches, number of capsules per plant, and yield). The performance of each of the sesame lines was analysed statistically. The standard deviation and coefficient of variation of each of the relevant traits were calculated separately for each treatment using an analysis of variance (ANOVA), and F and t-tests were used to assess the significance of differences between treatments [24, 25]. Using standard formulae, the heritability (broad sense) coefficients of various traits were calculated [26, 27].

To investigate the nature of the correlation between the contributory characteristics, all potential correlations among the four characteristics were calculated for each treatment/population at both phenotypic and genotypic levels. To evaluate whether a correlation coefficient was statistically significant, a t-test at (n—2) degrees of freedom was performed [28].

The frequency and magnitude of positive transgressive segregation for seed yield per plant were examined at the level of the individual plant. In the individual plant analysis mutants were compared to the highest-yielding individual of its better parent. By deducting the mean of positive transgressive segregates (PTS mean) from the better parent mean, the extent of transgression was determined [29].

## 3. Results

### 3.1. Variability and heredity in the $F_2M_2$ sesame generation

Table 3 shows the results of an ANOVA of means and heritability estimates for four key characteristics (plant height, the number of primary branches per plant, number of capsules per plant and the seed yield per plant) in the $F_2M_2$ sesame generation. There are significant differences in means among treatments for all characteristics except for the number of primary branches per sesame plant.

The significant heritability estimates for the different characteristics ranged from 55.70% for the sesame seed yield per plant to 81.30% for the number of capsules per plant. The heritability estimates were high for the number of capsules per plant; moderate for plant height and seed yield per plant, and low (and without significant difference) for the number of primary branches per plant. In Tables 4 and 5, an analysis of the variance of the standard deviation and the coefficient of variation of the different traits is shown.

The F test indicated significant variations between treatments in the standard deviation and coefficient of variability for all the examined characteristics, except the number of primary branches per plant. Considerable variation in these parameters among genotypes was expected because the parent materials involved were genotypically homogenous and segregating populations such as the $F_{2s}$ and $F_2M_{2s}$.

### 3.2. Comparison of characteristics in the $F_2M_2$ sesame generation

Tables 6 to 9 show the variability parameters of all the characteristics examined in the $F_2M_2$ sesame generation. Plant height variation was reduced in all mutants in $F_2{}^1M_2$ and $F_2{}^2M_2$ (Table 6).

The $^3F_2$ population had the highest range of variation (112.60–163.27), whereas the $^1F_2{}^1M_2$ population had the lowest range of variation (119.91–156.81). Only the $^2F_2{}^2M_2$ and $^3F_2{}^2M_2$ had significantly lower mean variation values, while the remaining three $F_2{}^1M_{2s}$ and one $^1F_2{}^2M_2$ had mean plant heights that were comparable to the appropriate control ($F_{2s}$) for this characteristic. Although the $^1F_2{}^1M_2$ and $^1F_2{}^2M_2$ populations exhibited a similar standard deviation in plant height as the control ($F_{2s}$) for this sesame characteristic, only the $^2F_2{}^2M_2$ and $^3F_2{}^2M_2$ populations had significantly smaller standard deviations than the control. For sesame

**Table 3. Analysis of variance of means and heritability estimates for four characteristics in the sesame $F_2M_2$ generation.**

| Sl. No. | Characteristics | Source of variation | | | Heritability Hbs (%) |
|---|---|---|---|---|---|
| | | Blocks df (2) | Treatments df (11) | Error df (22) | |
| 1. | Plant height (cm) | 55.135 | 72.680* | 23.091 | 67.70 |
| 2. | Primary branches per plant (no.) | 0.556 | 0.127 | 0.146 | 17.60 |
| 3. | Capsules per plant (no.) | 40.396 | 57.440** | 12.071 | 81.30 |
| 4. | Seed yield per plant (g) | 0.182 | 0.503* | 0.227 | 55.70 |

**Significant at the 1% level of probability and *significant at the 5% level of probability

**Table 4. Analysis of variance of standard deviation for four characteristics in the $F_2M_2$ sesame generation.**

| Sl. No. | Characteristics | Source of variation | | |
|---|---|---|---|---|
| | | Blocks df (2) | Treatments df (11) | Error df (22) |
| 1. | Plant height (cm) | 0.219 | 11.689** | 2.080 |
| 2. | Primary branches per plant (no.) | 0.054 | 0.028 | 0.033 |
| 3. | Capsules per plant (no.) | 14.657 | 89.854** | 4.385 |
| 4. | Seed yield per plant (g) | 0.000 | 0.809** | 0.130 |

**Significant at the 1% level of probability and *significant at the 5% level of probability

plant height the $^2F_2{}^2M_2$ and $^3F_2{}^2M_2$ populations both displayed lower mean and standard deviation values than the corresponding $^2F_2{}^1M_2$ and $^3F_2{}^1M_2$ (one cycle, EMS-treated) populations.

To ascertain which recurrent EMS treatment cycle of hybrids would be most beneficial to enlarge variability in subsequent generations, changes in the mean and standard deviation (expressed as percentages of the corresponding control) of several traits were also studied. Two cycles of mutagenesis treatment were applied to the hybrid materials, starting with $F_{1s}$. Hybrid plant height mean as a percentage of the control plant height mean decreased with increasing number of EMS treatments for up to two mutagenic cycles in the crosses Nirmala x Amrit and Prachi x Amrit, and increased from one to two mutagenic cycles in the cross Nirmala x Prachi (Fig 2a). For different $F_4M_4$ populations of the cross Nirmala x Prachi trends from one to two mutagenic cycles showed little change in mean plant height as a percentage of the control plant height. In the remaining mutant crosses s of Nirmala x Amrit and Prachi x Amrit, the mean standard deviation (as a percentage of the control plant standard deviation) decreased with increasing number of mutagenic cycles in the $F_2M_2$ populations (Fig 2b).

Table 7 illustrates characteristics of the number of primary branches per plant in the $F_2M_2$ generation. The $^1F_2$, $^3F_2$, $^2F_2{}^2M_2$, and $^3F_2{}^2M_2$ populations had the greatest ranges of variability in primary branches per plant, and the lowest range of variability was in the $^1F_2{}^1M_2$ population. In comparison to the corresponding controls ($F_{2s}$), two populations, $^2F_2{}^2M_2$ and $^3F_2{}^2M_2$, had a wider range of variation. All $F_2{}^1M_{2s}$ and $F_2{}^2M_{2s}$ had comparable means as controls. All $F_2{}^1M_{2s}$ and two $F_2{}^2M_{2s}$ populations, *i.e.* $^2F_2{}^2M_2$ and $^3F_2{}^2M_2$ (the two cycle EMS-treated population), had mean values similar to their respective controls ($F_{2s}$). For the number of primary branches per plant in sesame, only the $^1F_2{}^2M_2$ population had a significantly smaller standard deviation when compared to its corresponding control ($^1F_2$).

Fig 3a & 3b illustrate the mean and standard deviation of the number of primary branches per plant in the $F_2M_2$ population as a function of the number of EMS-treated cycles. From the one-cycle to the two-cycle mutant population in all crosses, the mean as a percentage of the

**Table 5. Analysis of variance of the coefficient of variation for four characteristics in the $F_2M_2$ sesame generation.**

| Sl. No. | Characteristics | Source of variation | | |
|---|---|---|---|---|
| | | Blocks df (2) | Treatments df (11) | Error df (22) |
| 1. | Plant height (cm) | 0.009 | 4.670** | 0.935 |
| 2. | Primary branches per plant (no.) | 3.801 | 16.620 | 17.187 |
| 3. | Capsules per plant (no.) | 8.986 | 131.431** | 6.811 |
| 4. | Seed yield per plant (g) | 1.730 | 66.847** | 14.929 |

(**Significant at the 1% level of probability and *significant at the 5% level of probability)

**Table 6. Sesame $F_2M_2$ generation plant height characteristics.**

| Treatments | Plant height range | Mean | Standard deviation |
|---|---|---|---|
| $V_1$ | 122.60–154.13 | 134.87 | 6.751 |
| $V_2$ | 119.39–152.39 | 134.76 | 5.857 |
| $V_3$ | 120.39–150.66 | 132.59 | 7.056 |
| $^1F_2$ | 118.90–163.00 | 138.77 | 9.380 |
| $^1F_2{}^1M_2$ | 119.91–156.81 | 136.11 | 8.187 |
| $^1F_2{}^2M_2$ | 116.52–159.52 | 137.76 | 8.335 |
| $^2F_2$ | 121.49–172.06 | 145.92 | 11.613 |
| $^2F_2{}^1M_2$ | 118.48–167.48 | 141.11 | 11.570 |
| $^2F_2{}^2M_2$ | 110.86–150.86 | 131.92* | 8.196* |
| $^3F_2$ | 112.60–163.27 | 139.00 | 10.257 |
| $^3F_2{}^1M_2$ | 122.14–160.28 | 143.74 | 9.771 |
| $^3F_2{}^2M_2$ | 117.63–156.40 | 129.17* | 6.229* |
| CD at 5% | - | 8.12 | 2.44 |
| SE (m) | - | 2.77 | 0.83 |
| CVe | - | 3.50 | 16.70 |

* Significantly different from the control. Treatments detail is available in Table 2

control treatment increased slightly with increasing EMS treatment across all $F_2M_{2s}$ populations.

Table 8 shows characteristics of the number of capsules per plant, which was highest (30.14–136.94) in the $^3F_2{}^2M_2$ population and lowest (35.68–111.28) in the $^2F_2$ population.

**Table 7. Sesame $F_2M_2$ generation characteristics of the number of primary branches per plant.**

| Treatment | Primary branches per plant range | Mean | Standard deviation |
|---|---|---|---|
| $V_1$ | 1.82–4.82 | 3.19 | 0.854 |
| $V_2$ | 1.81–5.18 | 3.21 | 0.843 |
| $V_3$ | 1.78–5.78 | 3.21 | 0.916 |
| $^1F_2$ | 1.37–7.37 | 3.40 | 1.221 |
| $^1F_2{}^1M_2$ | 1.87–4.90 | 3.23 | 0.894* |
| $^1F_2{}^2M_2$ | 1.76–6.76 | 3.42 | 1.005 |
| $^2F_2$ | 1.21–6.54 | 3.31 | 0.949 |
| $^2F_2{}^1M_2$ | 1.96–5.06 | 2.86 | 0.905 |
| $^2F_2{}^2M_2$ | 1.54–7.54 | 3.28 | 1.020 |
| $^3F_2$ | 1.02–7.02 | 3.66 | 1.000 |
| $^3F_2{}^1M_2$ | 1.87–5.87 | 3.13 | 0.990 |
| $^3F_2{}^2M_2$ | 1.54–7.54 | 3.54 | 1.091 |
| CD at 5% | - | 0.64 | 0.31 |
| SE (m) | - | 0.22 | 0.11 |
| CVe | - | 11.61 | 18.77 |

* Significantly different from the control. Treatments detail is available in Table 2. The population $^1F_2{}^1M_2$ and $^1F_2{}^2M_2$'s control is $^1F_2$. For populations $^2F_2{}^1M_2$ and $^2F_2{}^2M_2$, $^2F_2$ serves as the control. Populations $^3F_2{}^1M_2$ and $^3F_2{}^2M_2$'s control is $^3F_2$. Every population-$V_1$, $V_2$, $V_3$, $^1F_2$, $^1F_2{}^1M_2$, $^1F_2{}^2M_2$, $^2F_2$, $^2F_2{}^1M_2$, $^2F_2{}^2M_2$, $^3F_2$, $^3F_2{}^1M_2$, $^3F_2{}^2M_2$ signifies a particular type of treatment.

**Table 8. Sesame $F_2M_2$ generation characteristics of the number of capsules per plant.**

| Treatment | Number of capsules per plant range | Mean | Standard deviation |
|---|---|---|---|
| $V_1$ | 33.13–109.63 | 60.30 | 14.596 |
| $V_2$ | 34.86–100.06 | 62.79 | 14.986 |
| $V_3$ | 35.40–101.17 | 62.27 | 14.529 |
| $^1F_2$ | 23.00–119.97 | 62.87 | 18.932 |
| $^1F_2{}^1M_2$ | 32.84–126.38 | 68.54 | 22.642* |
| $^1F_2{}^2M_2$ | 33.38–127.24 | 71.64* | 27.223* |
| $^2F_2$ | 35.68–111.28 | 62.98 | 15.477 |
| $^2F_2{}^1M_2$ | 30.26–124.06 | 67.59 | 23.041* |
| $^2F_2{}^2M_2$ | 31.12–127.16 | 70.62* | 26.237* |
| $^3F_2$ | 31.94–148.94 | 65.84 | 22.546 |
| $^3F_2{}^1M_2$ | 26.72–128.49 | 65.59 | 27.716* |
| $^3F_2{}^2M_2$ | 30.14–136.94 | 74.58* | 28.147* |
| CD at 5% | - | 5.87 | 3.54 |
| SE (m) | - | 2.01 | 1.21 |
| CVe | - | 5.24 | 9.74 |

\* Significantly different from the control. Treatments detail is available in Table 2. The population $^1F_2{}^1M_2$ and $^1F_2{}^2M_2$'s control is $^1F_2$. For populations $^2F_2{}^1M_2$ and $^2F_2{}^2M_2$, $^2F_2$ serves as the control. Populations $^3F_2{}^1M_2$ and $^3F_2{}^2M_2$'s control is $^3F_2$. Every population-$V_1$, $V_2$, $V_3$, $^1F_2$, $^1F_2{}^1M_2$, $^1F_2{}^2M_2$, $^2F_2$, $^2F_2{}^1M_2$, $^2F_2{}^2M_2$, $^3F_2$, $^3F_2{}^1M_2$, $^3F_2{}^2M_2$ signifies a particular type of treatment.

**Table 9. Sesame $F_2M_2$ generation characteristics in seed yield per plant.**

| Treatment | Amount of seed yieldplant$^{-1}$ | Mean | Standard deviation |
|---|---|---|---|
| $V_1$ | 4.35–17.10 | 9.06 | 3.028 |
| $V_2$ | 4.20–15.90 | 8.97 | 2.759 |
| $V_3$ | 4.19–16.34 | 8.58 | 2.791 |
| $^1F_2$ | 3.58–17.39 | 8.87 | 2.773 |
| $^1F_2{}^1M_2$ | 5.36–19.12 | 9.77* | 3.209* |
| $^1F_2{}^2M_2$ | 2.76–20.39 | 9.54* | 4.234* |
| $^2F_2$ | 4.10–17.14 | 8.63 | 2.540 |
| $^2F_2{}^1M_2$ | 4.95–19.03 | 8.88 | 2.577 |
| $^2F_2{}^2M_2$ | 3.33–19.86 | 9.24* | 3.596* |
| $^3F_2$ | 4.94–23.63 | 9.80 | 3.493 |
| $^3F_2{}^1M_2$ | 5.26–16.81 | 9.09* | 2.642* |
| $^3F_2{}^2M_2$ | 3.16–20.37 | 8.80* | 3.172* |
| CD at 5% | - | 0.39 | 0.29 |
| SE (m) | - | 0.28 | 0.21 |
| CVe | - | 5.24 | 11.69 |

*Significantly different from the control. Treatments detail is available in Table 2. The population $^1F_2{}^1M_2$ and $^1F_2{}^2M_2$'s control is $^1F_2$. For populations $^2F_2{}^1M_2$ and $^2F_2{}^2M_2$, $^2F_2$ serves as the control. Populations $^3F_2{}^1M_2$ and $^3F_2{}^2M_2$'s control is $^3F_2$. Every population-$V_1$, $V_2$, $V_3$, $^1F_2$, $^1F_2{}^1M_2$, $^1F_2{}^2M_2$, $^2F_2$, $^2F_2{}^1M_2$, $^2F_2{}^2M_2$, $^3F_2$, $^3F_2{}^1M_2$, $^3F_2{}^2M_2$ signifies a particular type of treatment.

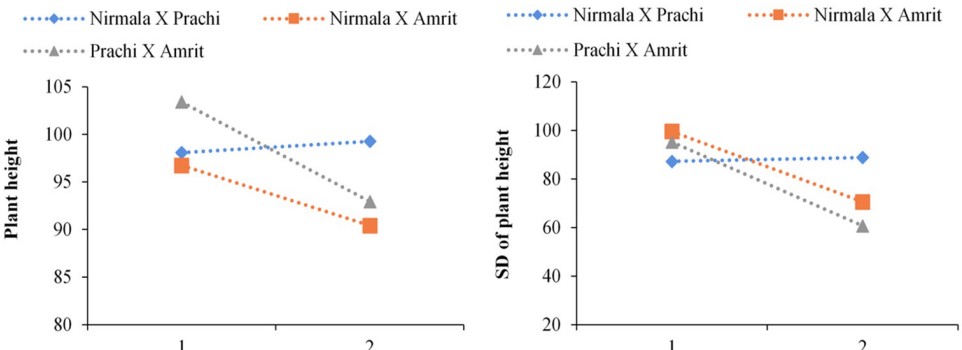

**Fig 2.** (**a**) Plant height means in $F_2M_2$ as a function of the number of EMS (ethyl methane sulfonate) treatment cycles; (**b**) Plant height standard deviations (SD) in $F_2M_2$ as a function of the number of EMS treatment cycles.

Except for the $^1F_2{}^1M_2$ and $^1F_2{}^2M_2$ populations, which had lower ranges than their corresponding controls, all the $F_2{}^1M_{2s}$ and $F_2{}^2M_{2s}$ populations had wider ranges of number of capsules per plant than their corresponding controls.

In terms of the number of capsules per sesame plant all $F_2{}^2M_{2s}$ populations had significantly higher means, and all $F_2{}^1M_{2s}$ populations had similar means, than their respective controls ($F_{2s}$), and all mutants showed a significantly higher standard deviation ($F_{2s}$). The $F_2{}^2M_{2s}$ population (two cycles mutated populations) had a higher standard deviation in the number of capsules per plant than the $F_2{}^1M_{2s}$ population (one cycle EMS treated populations), indicating that two cycles of EMS treatment may generate more variability than the corresponding controls or the one cycle of EMS treatment for this trait. Both the mean and standard deviation (as a percentage of the relevant control) of the number of capsules per plant increased from one to two cycles of mutagenesis (Fig 4a & 4b).

In comparison to the corresponding controls ($F_{2s}$), both the one cycle EMS-treated populations ($^1F_2{}^1M_2$ and $^2F_2{}^1M_2$) and the two cycle EMS-treated populations ($^1F_2{}^2M_2$ and $^2F_2{}^2M_2$) had a wider range of variance for the amount of seed yield per plant (Table 9). The one cycle mutated populations, populations $^1F_2{}^2M_2$ and $^2F_2{}^2M_2$, displayed a wider range of variance than their respective $F_2$ controls.

Compared to the $^3F_2$ population, the $^3F_2{}^1M_2$ and $^3F_2{}^2M_2$ populations had a narrower range of variation in seed yield. The $F_2{}^2M_2$ population of the Prachi x Amrit cross, however, had a

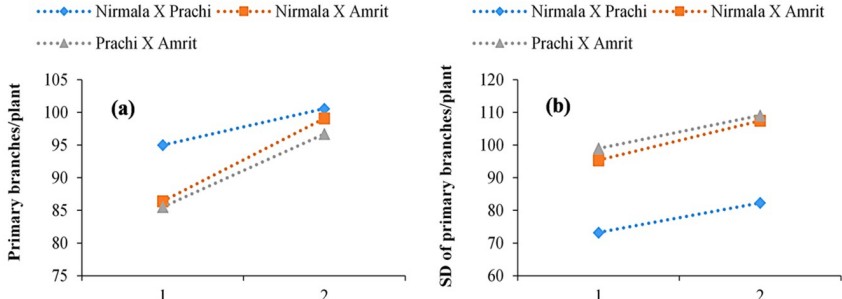

**Fig 3.** (**a**) Primary branches/plant means in $F_2M_2$ as a function of the number of EMS (ethyl methane sulfonate) treatment cycles; (**b**) Primary branches/plant standard deviations in $F_2M_2$ as a function of the number of EMS treatment cycles.

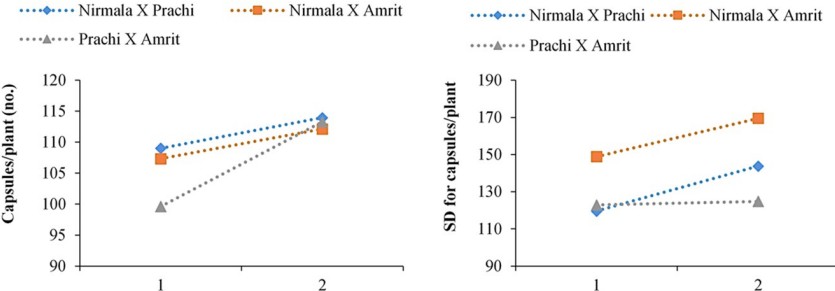

**Fig 4.** (**a**) Number of capsules/plant means in $F_2M_2$ as a function of the number of EMS (ethyl methane sulfonate) treatment cycles; (**b**) Number of capsules/ plant standard deviations in $F_2M_2$ as a function of the number of EMS treatment cycles.

wider range of variance than the $F_2{}^1M_2$ population of the Prachi x Amrit ($^3F_2{}^1M_2$) cross. Two $F_2{}^1M_2$s ($^1F_2{}^1M_2$ and $^3F_2{}^1M_2$) and all three $F_2{}^2M_2$ populations showed significant differences for mean sesame seed yields than their respective controls. The Nirmala x Prachi cross progenies $F_2{}^1M_2$ and $F_2{}^2M_2$ exhibited greater mean seed yields than the $^1F_2$ control. The offspring of the Nirmala x Amrit ($^2F_2{}^2M_2$) cross had a higher mean seed yield, while the $^2F_2{}^1M_2$ population had a mean seed yield comparable to that of $^2F_2$. Similarly, the Prachi x Amrit cross progeny $F_2{}^1M_2$ and $F_2{}^2M_2$ had lower mean seed yields than the $^3F_2$ control. Apart from the mutant population of the Nirmala x Amrit ($^2F_2{}^1M_2$) cross, all the mutant populations' standard deviations of seed yield differed significantly from those of the corresponding controls ($F_2$s).

The progenies of the $F_2M_2$ (Nirmala x Prachi) cross which were subjected to one and two cycles of EMS had significantly greater seed yield standard deviations than the corresponding controls. In comparison to the corresponding one-cycle mutant populations, the two-cycle mutated progenies, $^1F_2{}^2M_2$ and $^2F_2{}^2M_2$, had a greater standard deviation in seed yield, while the standard deviation was smaller in the $^3F_2{}^1M_2$ and $^3F_2{}^2M_2$ populations compared to the $F_2$ (Prachi x Amrit) control. The $^2F_2{}^1M_2$ population had a seed yield standard deviation comparable to that of the control $F_2$ (Nirmala x Amrit), however, the $^2F_2{}^2M_2$ population had a significantly higher standard deviation. Mean seed yield values (as a percentage of the respective control) increased between one to two cycles of mutagenic treatment in the Nirmala x Amrit cross, while they decreased between one to two cycles of mutagenic treatment in the Nirmala x

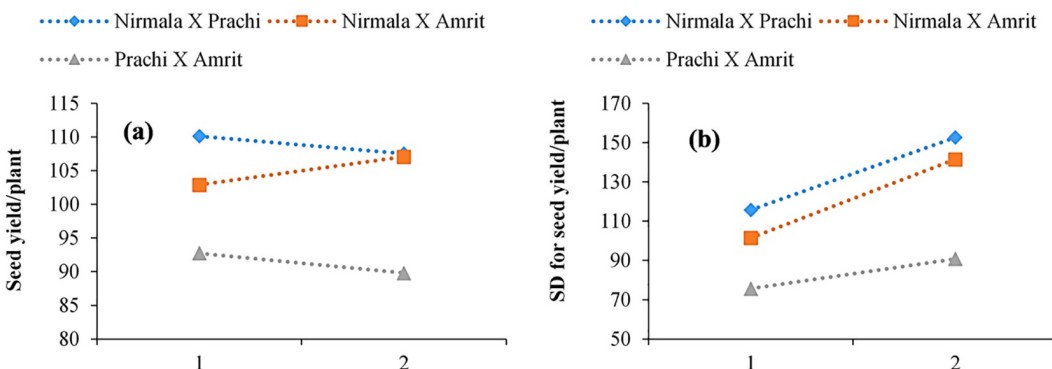

**Fig 5.** (**a**) Seed yield/plant means in $F_2M_2$ as a function of the number of EMS (ethyl methane sulfonate) treatment cycles; (**b**) Seed yield/ plant standard deviations in $F_2M_2$ as a function of the number of EMS treatment cycles.

Prachi and Prachi x Amrit crosses (Fig 5a). One to two cycles of mutagenic treatment resulted in an increase in variability for mean seed yield, but the amount of variability varied for different cross combinations, as shown in Fig 5b.

### 3.3. Correlatingsesame yield and its constituent traits in the $F_2M_2$generation

Phenotypic correlation in populations of different genotypes consists of both genotypic and environmental components. In contrast, the association between traits in populations with genotypically uniform individuals, such as pure-line varieties, is entirely due to environmental factors. A genotypic correlation is the outcome of genetic association, while an environmental correlation is the consequence of environmental factors which can influence how characteristics manifest. The nature and strength of the relationship between characteristics alters with alterations in the genotypic makeup of the population, and these changes depend on the type and extent of genetic alterations. In particular, in terms of the extent of any genetic change the relationship between characteristics may dissipate completely or partially if the associated characteristics' genetic changes are independent. Tables 10 and 11 illustrate the phenotypic and genotypic associations among different quantitative traits in the $F_2M_2$ sesame generation. At both phenotypic and genotypic levels, characteristics such as the number of primary branches and the number of capsules per plant have significant positive associations with sesame seed yield in all $F_2$ and $F_2M_2$ populations (i.e. after one and two cycles of EMS treatment). In contrast, plant height displayed a significant positive correlation with sesame seed yield per plant in all $F_2$, ${}^1F_2{}^1M_2$, ${}^2F_2{}^1M_2$, and ${}^3F_2{}^1M_2$ populations, while in the ${}^1F_2{}^2M_2$, ${}^2F_2{}^2M_2$, and ${}^3F_2{}^2M_2$ populations' plant height was not significantly associated with sesame seed yield at the phenotypic level. At the genotypic level in the mutant populations ${}^1F_2{}^1M_2$ and ${}^2F_2{}^1M_2$there were significant positive correlations between sesame plant height and seed yield.

### 3.4. Transgressive segregation at the level of individual plants in the $F_2M_2$ sesame generation

Table 12 shows the frequency and magnitude of positive transgressive variants for sesame seed yield per plant in the $F_{2s}$ and $F_2M_{2s}$ populations. All the $F_2M_2$and ${}^3F_2$populations generated

**Table 10. Phenotypic correlations between sesame yield and component characteristics in the $F_2M_2$ generation.**

| Treatment | $r_{12}$ | $r_{13}$ | $r_{14}$ | $r_{23}$ | $r_{24}$ | $r_{34}$ |
|---|---|---|---|---|---|---|
| $V_1$ | 0.125 | 0.200 | 0.236* | 0.753** | 0.743** | 0.912** |
| $V_2$ | 0.075 | 0.067 | 0.090 | 0.799** | 0.773** | 0.890** |
| $V_3$ | 0.162 | 0.246* | 0.280** | 0.657** | 0.624** | 0.924** |
| ${}^1F_2$ | 0.218* | 0.187 | 0.234* | 0.855** | 0.811** | 0.921** |
| ${}^1F_2{}^1M_2$ | 0.228* | 0.466** | 0.512** | 0.650** | 0.649** | 0.933** |
| ${}^1F_2{}^2M_2$ | -0.067 | 0.134 | 0.137 | 0.475** | 0.458** | 0.934** |
| ${}^2F_2$ | 0.290** | 0.336** | 0.264* | 0.652** | 0.518** | 0.878** |
| ${}^2F_2{}^1M_2$ | 0.273** | 0.447** | 0.512** | 0.536** | 0.533** | 0.869** |
| ${}^2F_2{}^2M_2$ | 0.100 | 0.192 | 0.180 | 0.659** | 0.614** | 0.882** |
| ${}^3F_2$ | 0.019 | 0.260* | 0.244* | 0.575** | 0.524** | 0.952** |
| ${}^3F_2{}^1M_2$ | 0.100 | 0.247* | 0.279** | 0.507** | 0.504** | 0.919** |
| ${}^3F_2{}^2M_2$ | 0.001 | 0.017 | 0.011 | 0.635** | 0.644** | 0.899** |

** Significant at 1% level of probability and *Significant at 5% level of probability, [1] plant height, [2] number of primary ranches/ plant, [3] number of capsules per plant, [4] seed yield per plant. Treatment details are shown in Table 2

**Table 11. Genotypic correlations among sesame yield and component characteristics in the $F_2M_2$ generation.**

| Treatment | $r_{12}$ | $r_{13}$ | $r_{14}$ | $r_{23}$ | $r_{24}$ | $r_{34}$ |
|---|---|---|---|---|---|---|
| $V_1$ | 0.180 | 0.169 | 0.188 | 0.791** | 0.766** | 0.891** |
| $V_2$ | 0.112 | 0.145 | 0.117 | 0.822** | 0.807** | 0.906** |
| $V_3$ | 0.124 | 0.463** | 0.470** | 0.693** | 0.671** | 0.943** |
| $^1F_2$ | 0.179 | 0.207 | 0.185 | 0.871** | 0.834** | 0.939** |
| $^1F_2{}^1M_2$ | 0.194 | 0.478** | 0.546** | 0.720** | 0.677** | 0.902** |
| $^1F_2{}^2M_2$ | -0.176 | 0.161 | 0.176 | 0.537** | 0.492** | 0.937** |
| $^2F_2$ | 0.205 | 0.468** | 0.193 | 0.663** | 0.573** | 0.889** |
| $^2F_2{}^1M_2$ | 0.194 | 0.493** | 0.634** | 0.587** | 0.578** | 0.895** |
| $^2F_2{}^2M_2$ | 0.187 | 0.203 | 0.138 | 0.710** | 0.690** | 0.902** |
| $^3F_2$ | 0.090 | 0.180 | 0.180 | 0.624** | 0.559** | 0.960** |
| $^3F_2{}^1M_2$ | 0.178 | 0.192 | 0.197 | 0.589** | 0.564** | 0.943** |
| $^3F_2{}^2M_2$ | 0.089 | 0.153 | 0.111 | 0.678** | 0.667** | 0.908** |

** Significant at 1% level of probability and *Significant at 5% level of probability, [1] plant height, [2] number of primary branches per plant, [3] number of capsules per plant, [4] seed yield/ plant. Treatment details are shown in Table 2

transgressive segregants that outperformed the seed yield of the highest-yielding plant of their better parent. Of 90 plants, there were five such plants identified in the $^3F_2$ (Prachi x Amrit) population. The frequency of these transgressive segregants in the $F_2M_2$ population ranged from one in the one-cycle mutated population of $^1F_2{}^1M_2$ (Nirmala x Prachi), $^2F_2{}^1M_2$ (Nirmala x Amrit) and $^3F_2{}^1M_2$ (Prachi x Amrit) to eight in the two-cycle mutated population of $^1F_2{}^2M_2$ (Nirmala x Prachi). The mean sesame seed yield per transgressive plant (PTS mean) in $^3F_2$ was 18.81 g/plant. The PTS means of the $F_2M_{2s}$ ranged from 16.81 g/plant in the one-cycle mutant population $^3F_2{}^1M_2$ to 19.64 g/plant in the two-cycle mutated population $^3F_2{}^2M_2$. The $^3F_2$ (Prachi x Amrit) population had an average transgression of 10.23 g/plant (PTS mean—better parent mean). In the $F_2M_2$ populations the average transgression ranged from 8.23 g/plant in the one-cycle EMS-treated Prachi x Amrit population to 11.06 g/plant in the two-cycle EMS-treated Prachi x Amrit population. Considering the three parameters of transgressive variation at the same time indicates that multiple $F_2M_2$ populations offer ample scope for further yield improvement in sesame.

**Table 12. Frequency and magnitude of positive transgressions of sesame seed yield in the $F_2M_2$ generation after recurrent EMS (ethyl methane sulfonate) treatment of hybrids from three sesame varieties.**

| Treatment | Positive transgression range | Positive transgression mean (g/plant) | Better parent range | Better parent mean (g/plant) | Frequency of PTS | Mean of PTS (g/plant) | Average transgression |
|---|---|---|---|---|---|---|---|
| $^1F_2$ | 3.58–17.39 | 8.87 | 4.35–18.10 | 9.06 | - | - | - |
| $^1F_2{}^1M_2$ | 5.36–19.12 | 9.77 | 4.35–18.10 | 9.06 | 1 | 19.12 | 10.06 |
| $^1F_2{}^2M_2$ | 2.76–20.39 | 9.54 | 4.35–18.10 | 9.06 | 8 | 18.97 | 9.91 |
| $^2F_2$ | 4.10–17.14 | 8.63 | 4.35–18.10 | 9.06 | - | - | - |
| $^2F_2{}^1M_2$ | 4.95–19.03 | 8.88 | 4.35–18.10 | 9.06 | 1 | 19.03 | 9.97 |
| $^2F_2{}^2M_2$ | 3.33–19.86 | 9.24 | 4.35–18.10 | 9.06 | 3 | 18.84 | 9.78 |
| $^3F_2$ | 4.94–23.63 | 9.80 | 4.19–16.34 | 8.58 | 5 | 18.81 | 10.23 |
| $^3F_2{}^1M_2$ | 5.26–16.81 | 9.09 | 4.19–16.34 | 8.58 | 1 | 16.81 | 8.23 |
| $^3F_2{}^2M_2$ | 3.16–20.37 | 8.80 | 4.19–16.34 | 8.58 | 3 | 19.64 | 11.06 |

Treatment detailsare shown in Table 2; PTS, positive transgressive segregates; EMS, ethyl methane sulfonate.

## 4. Discussion

### 4.1. Sesame $F_2M_2$ generation mutagenic response

ANOVAs of the mean, standard deviation, and coefficients of variation of sesame yield and key yield-attributing traits in the $F_2M_2$ generation indicated significant treatment shifts, with the exception of the number of primary branches (Tables 2 to 5). The treatment differences with respect to the standard deviation and coefficient of variation may be the result of EMS's mutagenic and/or physiological effects. The heritability (broad sense) estimates for different characteristics ranged from 17.60% for the number of primary branches per sesame plant to 81.30% for the number of capsules per sesame plant (Table 3). The heritability estimates were high for the number of capsules per plant, moderate for plant height and seed yield per plant, and low for the number of primary branches per plant. These findings are in broad conformity with those reported elsewhere [30–32]. Thus, chances for selection to be operative were high for all the studied characteristics except for the number of primary branches per plant.

All the $F_2M_2$ generation mutants from the various crosses had decreased plant height (Table 6). All the one-cycle EMS-treated mutants had plant heights that were comparable to their respective $F_2$ populations in terms of mean and variability. In comparison to their respective controls ($F_2$), the mutants $^2F_2{}^2M_2$ and $^3F_2{}^2M_2$ showed significantly lower means for this characteristic, and other mutants were similar to their respective controls ($F_2$). Also, compared to their respective controls ($F_2$), the mutants $^2F_2{}^2M_2$ and $^3F_2{}^2M_2$ had considerably reduced variability. For plant height in sesame, the remaining mutants ($F_2{}^1M_{2s}$ and $^1F_2{}^2M_2$) showed variability that was comparable to that of their respective controls ($F_2$). Overall the $^2F_2{}^2M_2$ and $^3F_2{}^2M_2$ populations had lower mean and variability than comparable one-cycle mutant populations for this characteristic.

In general populations which had undertaken two cycles of EMS treatment had lower mean and variability in sesame plant height than corresponding populations that underwent one cycle of mutation. In general, populations treated with EMS for two cycles showed a substantial negative change in both the mean and variability of plant height, whereas populations treated with EMS for one cycle did not demonstrate significant changes in plant height mean and variation. The reductions in plant height mean and variability in the current study could be caused by a sufficient frequency of mutations with harmful effects [33].

In terms of the number of primary branches per sesame plant, the $^2F_2{}^2M_2$ population of Nirmala x Amrit and the $^3F_2{}^2M_2$ population of Prachi x Amrit had greater ranges of variation than their respective $F_2$ populations (Table 7). All the $F_2M_2$ populations had similar means for the number of primary branches per plant relative to their respective control $F_2$ populations, while only the $^1F_2{}^1M_2$ population had significantly lower variability than its respective control population. The number of primary branches per plant had very little induced variability, with no significant change in the observed mean value after either one or two cycles of EMS-treated populations compared to the respective control populations.

All the $F_2{}^1M_2$ and $F_2{}^2M_2$ populations had a greater range of variation relative to their control populations in terms of the number of seed capsules per sesame plant, with the exception of the $^1F_2{}^1M_2$ and $^1F_2{}^2M_2$ populations (Table 8). When compared to their respective control $F_2$ populations, all the recurrent EMS-treated $F_2M_2$ populations had higher values for the mean and increased variability in the number of seed capsules per plant, whereas the one-cycle EMS-treated $F_2M_2$ populations were similar to the control populations. In general, the populations which had received two cycles of EMS-treatment had significantly greater variability than the one cycle of EMS-treated populations for the number of capsules per plant, and the two-cycle EMS-treated populations $^1F_2{}^2M_2$ and $^2F_2{}^2M_2$ had significantly greater variability than their respective one-cycle mutant populations. For the number of seed capsules per plant,

two cycles of EMS treatment was more effective in enlarging variability and positively adjusting the mean. The positive shift in mean values with increased variability occurred more frequently under polygenic mutation and with a higher frequency of mutations with positive effects.

In terms of the seed yield per sesame plant, two single-cycle EMS-treated populations ($^1F_2^1M_2$ and $^2F_2^1M_2$) and two double-cycle EMS-treated populations ($^1F_2^2M_2$ and $^2F_2^2M_2$) showed a greater range of variation relative to their respective controls ($F_2$) (Table 9). When compared to their respective control populations $^3F_2$, the $^3F_2^1M_2$ and $^3F_2^2M_2$ populations had a lower range of variation. Of the six $F_2M_{2s}$ populations achieved subsequent to one or two cycles of EMS treatment, two populations had decreased means, one population had a similar mean, and three populations had increased means relative to their respective control ($F_2$) populations. Similarly, out of six $F_2M_{2s}$, one had a decreased mean, one had a similar mean, and four had an increased mean for seed yield per plant compared to the respective $F_2$ populations. The one- and two-cycle EMS-treated populations of $F_2M_2$ (Nirmala x Prachi) and $F_2M_2$ (Nirmala x Amrit) had significantly greater variability than their respective controls. In general, the two-cycle EMS-treated populations $^1F_2^2M_2$ and $^2F_2^2M_2$ had greater variability in seed yield per plant than the respective one-cycle EMS-treated populations. Recurrently mutagenized populations had greater variability than corresponding one-cycle EMS-treated populations in terms of seed yield per sesame plant.

Overall, $F_2M_2$ populations resulted in a negative shift in mean plant height, and a positive shift in mean capsule number and seed yield in mutants compared to the respective control populations. Similar results have been noted elsewhere for plant height and seed yield [34], for the number of capsules per plant and seed yield per plant [35], for capsule number [36], for seed yield [37], and for capsule number and seed yield [38].

The analysis of the $F_2M_2$ populations indicated an enhanced variability in number of seed capsules per plant and in sesame seed yield but decreased variability for plant height in mutant populations compared to the respective controls. Induced micro-mutations are responsible for the increase or decrease in the variability in these traits, and the segregation and recombination of polygenes in hybrid populations' results in cumulative effects. Similar findings have been noted elsewhere for seed yield, plant height, capsules, and seed yield [39–41]. A wide range of variability for different traits in $F_2M_2$ populations has been observed by [19, 42], who concluded that $F_2M_2$ populations have the potential to offer a greater scope for evolving high-yielding sesame varieties than either $F_2$ or $M_2$ populations.

## 4.2. Character correlation in the sesame $F_2M_2$ generation

The phenotypic association between traits in genotypically diverse populations results from both genetic and environmental factors. The association between characters may alter in response to changes in a population's genotypic makeup. The relationship between characteristics may dissipate either completely or partially as a result of random, independent mutations in polygenes that affect several characteristics. For all populations of the $F_2M_2$ generation (both untreated and treated), phenotypic and genotypic correlations among four characteristics were assessed and the results are shown in Tables 10 and 11.

The number of primary branches per sesame plant and the number of capsules per sesame plant both showed significant positive association with the number of seeds per sesame plant in all $F_2$ and $F_2M_2$ populations at both phenotypic and genotypic levels. The number of capsules per plant and the seed yield per plant in sesame genotypes was positively correlated, which was also observed by [43–47]. A positive correlation between plant height, branch number, and capsule number per plant with sesame seed yield per plant was also observed by [46–

51]. Sesame plant height showed a significant positive correlation with seed yield per sesame plant in all the $F_2$ crosses and mutants (e.g. $^1F_2{}^1M_2$, $^2F_2{}^1M_2$ and $^3F_2{}^1M_2$, etc) whereas in the $^1F_2{}^2M_2$, $^2F_2{}^2M_2$ and $^3F_2{}^2M_2$ mutants the association was positive but not significant at the phenotypic level. In contrast at the genotypic level there were significant positive correlations between sesame plant height and seed yield per sesame plant in the mutant populations $^1F_2{}^1M_2$ and $^2F_2{}^1M_2$. A non-significant and positive association between plant height and the number of primary branches per plant with seed yield per plant in sesame genotypes was reported by [43, 45, 52]. Changes in the associations between yield and its component characteristics in $F_2M_2$ mutant populations compared to corresponding controls suggests that EMS treatments induced significant genetic alterations. Independent polygenic mutations and the increased recombination of polygenes influencing several traits are likely to have caused the observed association changes.

### 4.3. Individual plant transgressive segregation in the $F_2M_2$ sesame generation

Analysis of positive transgressive segregation for yield was undertaken at the individual plant level in the $F_2M_2$ generation, taking into account the frequency of transgressive segregants and the magnitude of transgression. The frequency of positive transgressive variants and the magnitude of average transgression of yield in the $F_2$ and $F_2M_2$ populations are shown in Table 12. The $^3F_2$ population and all mutant $F_2M_2$ populations produced transgressive segregants resulting in higher yields than the highest-yielding plants of their respective better parents. The prevalence of such transgressive segregants among the $F_2M_2$ populations ranged from one in the $^1F_2{}^1M_2$ and $^3F_2{}^1M_2$ populations to eight in the $^1F_2{}^2M_2$ population.

The mean yield of the $F_2M_2$ populations ranged from 16.81 g/plant in the one-EMS treated Prachi x Amrit ($^3F_2{}^1M_2$) population to 19.64 g/plant in the two-EMS treated Prachi x Amrit ($^3F_2{}^2M_2$) population. In the $F_2M_2$ populations, the average transgression rates ranged from 8.23 g/plant in the $F_2{}^1M_2$ Prachi x Amrit population to 11.06 g/plant in the $F_2{}^2M_2$ Prachi x Amrit population. A simultaneous consideration of the three parameters of transgressive variation indicates that many $F_2M_2$ populations have similar scope for further selection of yield characteristics and that the prospects of improving on parent materials were high in all cases. Most $F_2M_2$ populations had a higher frequency of transgressive segregants and a higher average transgression, as well as more potential for yield selection than their respective control ($F_2$) populations. The positive transgressive segregants for seed yield per plant during the evaluation of several $F_2$ populations of sesame have also been reported by [42, 48], while positive transgressive segregants for seed yield per plant in $F_2M_2$ populations of sesame were reported by [19, 42] who observed the importance of combining hybridization and mutagenesis to produce superior and greater frequencies of transgressive segregants for seed yield per plant in sesame.

## 5. Conclusions

The recurrent EMS treatment of inter-varietal hybrids increased variability in segregating generations of sesame, and this effect increased with the number of EMS treatment cycles, particularly in terms of the seed capsule number per sesame plant, and the seed yield per sesame plant. As indicated by heritability estimates the ability to select positive characteristics was high for all the characteristics examined except the number of primary branches per plant. At both phenotypic and genotypic levels, the number of primary branches per sesame plant and the number of capsules per sesame plant were significantly positively correlated with sesame seed yield per plant in all $F_2M_2$ populations. Independent mutations of quantitative traits and

better recombination of the genes controlling these traits are likely to have caused these association changes. Two cycles of mutagenic treatment resulted in increased variability, higher transgressive segregates, and improved mean and average transgression rates for sesame seed yield per plant compared to either the corresponding control or to one cycle of mutagenic treatment. These findings demonstrate the value of conducting two EMS treatment cycles to improve key characteristics in sesame: cumulative variation in hybrids is caused by segregation and artificial micromutations. Furthermore, the additionally variability created through the extended hybridization could be utilized for further breeding work, and the creation of even better seed yield segregants would be a further advance for selection in order to develop ever more superior varieties of sesame.

## Supporting information

**S1 Data.**
(XLSX)

## Acknowledgments

The authors acknowledge the Department of Genetics and Plant Breeding, College of Agriculture, Odisha University of Agriculture and Technology, Bhubaneswar 751003, Odisha, India.

## Author Contributions

**Conceptualization:** Rajesh Kumar Kar, Tapash Kumar Mishra, Dibyabharati Sahu, Subhashree Das, Deepak Kumar Swain, Srikrushna Behera, Aditya Kiran Padhiary, Sarthak Pattanayak, S. P. Monalisa, Ritu Kumari Pandey, Poonam Preeti Pradhan, Debendra Nath Sarangi, Mihir Ranjan Mohanty, Biswajit Lenka, Lingaraj Dip, Anannya Jena, Uma Pradhan, Siba Prasad Mishra, Manas Kumar Patel, Rashmi Prabha Mishra.

**Data curation:** Rajesh Kumar Kar, Ahmed Gaber, Akbar Hossain.

**Formal analysis:** Rajesh Kumar Kar, Ahmed Gaber, Akbar Hossain.

**Funding acquisition:** Ahmed Gaber, Akbar Hossain.

**Investigation:** Rajesh Kumar Kar, Banshidhar Pradhan, Dibyabharati Sahu, Deepak Kumar Swain, Sarthak Pattanayak, S. P. Monalisa, Ritu Kumari Pandey, Biswajit Lenka, Anannya Jena, Uma Pradhan, Siba Prasad Mishra, Rashmi Prabha Mishra.

**Methodology:** Rajesh Kumar Kar, Tapash Kumar Mishra, Banshidhar Pradhan, Dibyabharati Sahu, Subhashree Das, Srikrushna Behera, Sarthak Pattanayak, S. P. Monalisa, Ritu Kumari Pandey, Mihir Ranjan Mohanty, Biswajit Lenka, Lingaraj Dip, Anannya Jena, Uma Pradhan, Siba Prasad Mishra, Manas Kumar Patel, Rashmi Prabha Mishra.

**Resources:** Tapash Kumar Mishra, Banshidhar Pradhan.

**Software:** Rajesh Kumar Kar.

**Supervision:** Tapash Kumar Mishra, Banshidhar Pradhan.

**Validation:** Rajesh Kumar Kar, Banshidhar Pradhan, Dibyabharati Sahu, Subhashree Das, Deepak Kumar Swain, Srikrushna Behera, Aditya Kiran Padhiary, Sarthak Pattanayak, S. P. Monalisa, Ritu Kumari Pandey, Poonam Preeti Pradhan, Debendra Nath Sarangi, Mihir Ranjan Mohanty, Biswajit Lenka, Lingaraj Dip, Anannya Jena, Uma Pradhan, Siba Prasad Mishra, Manas Kumar Patel, Rashmi Prabha Mishra.

**Visualization:** Rajesh Kumar Kar, Tapash Kumar Mishra, Banshidhar Pradhan, Subhashree Das, Deepak Kumar Swain, Srikrushna Behera, Aditya Kiran Padhiary, Sarthak Pattanayak, S. P. Monalisa, Ritu Kumari Pandey, Poonam Preeti Pradhan, Debendra Nath Sarangi, Mihir Ranjan Mohanty, Biswajit Lenka, Lingaraj Dip, Anannya Jena, Uma Pradhan, Siba Prasad Mishra, Manas Kumar Patel, Rashmi Prabha Mishra.

**Writing – original draft:** Rajesh Kumar Kar, Tapash Kumar Mishra, Banshidhar Pradhan, Dibyabharati Sahu, Subhashree Das, Deepak Kumar Swain, Srikrushna Behera, Aditya Kiran Padhiary, Sarthak Pattanayak, S. P. Monalisa, Ritu Kumari Pandey, Poonam Preeti Pradhan, Debendra Nath Sarangi, Mihir Ranjan Mohanty, Biswajit Lenka, Lingaraj Dip, Anannya Jena, Uma Pradhan, Siba Prasad Mishra, Manas Kumar Patel, Rashmi Prabha Mishra, Akbar Hossain.

**Writing – review & editing:** Ahmed Gaber, Akbar Hossain.

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
