## [Decision Letter · Decision Letter 0]

10 May 2023

PONE-D-23-11823

Effectiveness of Repeated Mutagenesis of Sesame Crosses for Enhancing Polygenic Variability in F2M2 Generation

PLOS ONE

Dear Dr. Hossain,

Thank you for submitting your manuscript to PLOS ONE. After careful consideration, we feel that it has merit but does not fully meet PLOS ONE’s publication criteria as it currently stands. Therefore, we invite you to submit a revised version of the manuscript that addresses the points raised during the review process.

We look forward to receiving your revised manuscript.

Kind regards,

Mojtaba Kordrostami, Ph.D.

Academic Editor

PLOS ONE

Journal Requirements:

    "The authors acknowledge the Department of Genetics and Plant Breeding, College of Agriculture, Odisha University of Agriculture and Technology, Bhubaneswar-751003, Odisha, India and the Taif University, Taif, Saudi Arabia for providing all facilities during conducting the study."

    "This research work was supported by the Department of Genetics and Plant Breeding, College of Agriculture, Odisha University of Agriculture and Technology, Bhubaneswar-751003, Odisha, India. The study was also partially funded by the Taif University Researchers Supporting Project number (TURSP - 2020/39), Taif University, Taif, Saudi Arabia. The funders had no role in study design, data collection and analysis, decision to publish, or preparation of the manuscript."

Additional Editor Comments:

Dear Authors,

Thank you for submitting your manuscript to PLOS ONE. We have received two expert reviews, which have provided valuable feedback on your work. We have carefully evaluated your manuscript and believe that it has potential to be published in PLOS ONE after major revisions. Please carefully address the comments and suggestions provided by the reviewers and resubmit your manuscript along with a point-by-point response to each comment.

Reviewer 1 has provided detailed feedback on several aspects of your manuscript. They have suggested adding new references to the Introduction, providing a comparison of your results with new references in the Discussion section, and clarifying the materials mentioned in line 156. They have also suggested that you provide a brief summary of the results related to yield stability parameters in India in line 132 and explain how you derived resilience for climatic change for the mentioned lines. In addition, they have provided several specific comments on certain lines in your manuscript that would benefit from revision. These include line 143, where they suggest specifying which lines are maternal and explaining why you didn't use reciprocal crosses, and line 287, where they suggest clarifying what is meant by "determine the control (?) treatment."

Reviewer 1 has also suggested that you calculate genetic correlation instead of phenotypic correlation in Table 10 and provide an explanation for the low values such as r13=0.26. They have recommended presenting the ANOVA analysis for the raw data of the studied characters rather than just the CV or SD in Tables 3, 4, and 5. Additionally, they suggest providing an explanation for not using the cross of Nirmala and Prachi and revising line 154 to read "other workers' work" instead of "alter workers to works." Lastly, they suggest adding a reference and formula for the transgressive segregation calculation in line 209 and calculating and presenting the heritability related to each character in the ANOVA tables.

Reviewer 2 has provided similar feedback and has also suggested that you address the comments provided by Reviewer 1. They have specifically commented on line 132 and requested a brief presentation of the results related to yield stability parameters in India or an explanation of how you derived resilience for climatic change for the mentioned lines. In addition, they have suggested specifying which lines are maternal in line 143 and explaining why you didn't use reciprocal crosses. They have also commented on the low values such as r13=0.26 in Table 10 and recommended calculating genetic correlation instead of phenotypic correlation. They have also recommended presenting the ANOVA analysis for the raw data of the studied characters rather than just the CV or SD in Tables 3, 4, and 5. Lastly, they have requested adding a reference for the transgressive segregation calculation and formula in line 209 and calculating and presenting the heritability related to each character in the ANOVA tables.

We encourage you to carefully consider all of the comments provided by the reviewers and make the necessary revisions to your manuscript. Please submit a point-by-point response to each comment along with your revised manuscript. We look forward to receiving your revised manuscript and are available to assist you in any way we can.

Sincerely,

Mojtaba Kordrostami

Editor, PLOS ONE

Reviewers' comments:

Reviewer's Responses to Questions

**Comments to the Author**

1. Is the manuscript technically sound, and do the data support the conclusions?

Reviewer #1: Partly

Reviewer #2: Yes

2. Has the statistical analysis been performed appropriately and rigorously? 

Reviewer #1: No

Reviewer #2: Yes

3. Have the authors made all data underlying the findings in their manuscript fully available?

Reviewer #1: Yes

Reviewer #2: Yes

4. Is the manuscript presented in an intelligible fashion and written in standard English?

Reviewer #1: Yes

Reviewer #2: Yes

5. Review Comments to the Author

Reviewer #1: Please refer to reviewer comments:

In line 132: are these lines were studied based on yield stability parameters in India? If yes, please present a brief results. If no, how can authors find resilience for climatic change for mentioned lines…

In line 143: which lines are maternal? Why the authors dont used reciprocal crosses ? whether it be could influenced the mutagenesis results?

In line 143: why authors did not used cross of Nirmala and Prachi?

In line 154: alter workers to works

In line 155: previous research (must be mentioned?????)

In line 156: …. Materials (seed?plant?)

In line 187: Randomized Complete Block design is correct since each block have all treatments.

In line 199: replace "observed' with "measured'

I line 287: determine the control (?) treatment

In Table 10: highly recommended to calculate genetic correlation instead of phenotypic correlation

In Table 10: the r13=0.26 or other such low values are significant. How authors explain this item?

In Table 3, 4 and 5: please present the ANOVA analysis for raw data of studied characters (NOT CV or SD)

In line 209: add reference for transgressive segregation calculation and formula

In ANOVA tables: calculate and present the heritability related to each character

Reviewer #2: Please add some new references to the Introduction

Please compare your results with new references in the discussion section

In line 132, it would be helpful to provide a brief summary of the results related to yield stability parameters in India, if they were studied. If not, it would be useful for the authors to explain how they derived the resilience for climatic change for the mentioned lines.

Regarding line 143, it would be beneficial for the authors to specify which lines are maternal and explain why they didn't use reciprocal crosses. Additionally, it would be useful to discuss whether using reciprocal crosses could have influenced the mutagenesis results. The authors could also provide an explanation for not using the cross of Nirmala and Prachi.

Please consider revising line 154 to read "other workers' work" instead of "alter workers to works."

In line 155, it would be helpful for the authors to provide references for previous research in this area.

Please clarify whether the materials mentioned in line 156 refer to seeds or plants.

In line 187, it is correct to use Randomized Complete Block design since each block contains all treatments.

Please replace "observed" with "measured" in line 199.

In line 287, please clarify what is meant by "determine the control (?) treatment."

Highly recommend calculating genetic correlation instead of phenotypic correlation in Table 10.

Please provide an explanation for the low values such as r13=0.26 in Table 10.

For Tables 3, 4, and 5, it would be beneficial to present the ANOVA analysis for the raw data of the studied characters, rather than just the CV or SD.

Please add a reference and formula for the transgressive segregation calculation in line 209.

It would be helpful to calculate and present the heritability related to each character in the ANOVA tables.

6. PLOS authors have the option to publish the peer review history of their article (what does this mean?). If published, this will include your full peer review and any attached files.

Reviewer #1: No

Reviewer #2: No

---

## [Author Response · Author response to Decision Letter 0]

10 Jul 2023

Response to editorial comments

Dear Editor

Thank you so much for your comments to improve the language of the article. We are happy to inform you that we have been able to improve the english language of the article by a native english speaker. If unfortunately, we have overlooked any aspects kindly let us know and we will rectify that immediately. Please note that all edits are shown in track change mode in the text of the article.

We look forward to the editorial decision!

Sincerely 

Akbar Hossain, Ph.D

http://orcid.org/0000-0003-0264-2712

---

## [Decision Letter · Decision Letter 1]

27 Jul 2023

Effectiveness of Repeated Mutagenesis of Sesame Crosses for Enhancing Polygenic Variability in F2M2 Generation

PONE-D-23-11823R1

Dear Dr. Hossain,

We’re pleased to inform you that your manuscript has been judged scientifically suitable for publication and will be formally accepted for publication once it meets all outstanding technical requirements.

Kind regards,

Mojtaba Kordrostami, Ph.D.

Academic Editor

PLOS ONE

Additional Editor Comments (optional):

The manuscript can be accepted now

Reviewers' comments:

Reviewer's Responses to Questions

**Comments to the Author**

1. If the authors have adequately addressed your comments raised in a previous round of review and you feel that this manuscript is now acceptable for publication, you may indicate that here to bypass the “Comments to the Author” section, enter your conflict of interest statement in the “Confidential to Editor” section, and submit your "Accept" recommendation.

Reviewer #1: All comments have been addressed

Reviewer #2: All comments have been addressed

2. Is the manuscript technically sound, and do the data support the conclusions?

Reviewer #1: No

Reviewer #2: Yes

3. Has the statistical analysis been performed appropriately and rigorously? 

Reviewer #1: Yes

Reviewer #2: Yes

4. Have the authors made all data underlying the findings in their manuscript fully available?

Reviewer #1: Yes

Reviewer #2: Yes

5. Is the manuscript presented in an intelligible fashion and written in standard English?

Reviewer #1: Yes

Reviewer #2: No

6. Review Comments to the Author

Reviewer #1: (No Response)

Reviewer #2: (No Response)

7. PLOS authors have the option to publish the peer review history of their article (what does this mean?). If published, this will include your full peer review and any attached files.

Reviewer #1: No

Reviewer #2: No

---

## [Editor Report · Acceptance letter]

31 Jul 2023

PONE-D-23-11823R1 

Effectiveness of Repeated Mutagenesis of Sesame Crosses for Enhancing Polygenic Variability in F2M2 Generation 

Dear Dr. Hossain:

I'm pleased to inform you that your manuscript has been deemed suitable for publication in PLOS ONE. Congratulations! Your manuscript is now with our production department. 

Kind regards, 

on behalf of

Dr. Mojtaba Kordrostami 

Academic Editor

PLOS ONE